# Optimizing Federated Learning Client Selection via Multi-Objective Contextual Bandits

## Abstract

In the rapidly evolving field of Machine Learning (ML), Federated Learning (FL) emerges as an innovative approach for training models across distributed devices without centralizing raw data. However, FL faces significant challenges due to the heterogeneous nature of client devices, leading to non-IID data distributions and various resource constraints. Moreover, the inherent bandwidth limitations in decentralized settings necessitate the efficient use of both network and energy resources. Energy efficiency—i.e., minimizing energy consumed per unit time of training—not only reduces battery strain but also cuts down on unnecessary data transmissions. This reduction in energy usage not only improves network efficiency but also contributes to environmental sustainability. To address these challenges, we introduce a novel solution, Pareto Contextual Zooming for Federated Learning (PCZFL), which treats the client selection problem in FL as a multi-objective bandit problem. Our method focuses on optimizing both global accuracy and energy efficiency in parallel. By dynamically adjusting client selection based on real-time accuracy and energy context, the proposed solution ensures effective participation while minimizing energy consumption. In addition, we provide theoretical analysis on both the regret bound and time complexity of our method. Extensive experiments on CIFAR-10 demonstrate that PCZFL achieves a **4.7% improvement in global accuracy** with **37.3% lower energy cost** compared to Pow-d, the second-best in global accuracy, and delivers a **9.5% accuracy gain** while reducing **energy consumption by 30.9%** relative to NCCB, the second-best in energy efficiency. On FMNIST, PCZFL further achieves a **3.2% improvement in global accuracy** and a **21.4% reduction in energy cost** relative to Pow-d, while delivering a **3.7% accuracy gain** and a **30.8% reduction in energy consumption** relative to NCCB.

## 1 Introduction

The increasing integration of Federated Learning (FL) (McMahan et al., 2016) within the Internet of Things (IoT) landscape marks a significant shift in decentralized data processing. Developed to tackle the inherent challenges of traditional centralized learning models, FL enables a multitude of devices, such as IoT sensors and smartphones, to collaboratively train a global model while keeping all personal data localized. This method effectively addresses several critical issues (Dubey & Kumar, 2025): it enhances data security by adhering to privacy regulations that restrict data sharing; it leverages the computational power of massive scale of heterogeneity devices; it reduces bandwidth usage by minimizing data transmission, requiring only the exchange of model updates between the client devices and a central server. Such model updates are conducted over finite communication rounds, where the server aggregates clients' local model parameters to obtain an improved global model.

Despite its advantages, FL also faces several challenges that can impact the efficiency and effectiveness of the training process. One significant challenge in FL is managing the limited bandwidth when the size of the participating clients is large (Zhang et al., 2023). Strategically selecting a subset of clients to participate in model training, without compromising overall model performance, is a promising solution to address resource contention on networks. As such, prioritizing devices based on their connectivity and data capabilities becomes crucial. Another challenge FL faces is the variability in both the quantity and quality of datasets

across devices, which often leads to significantly different local models (Lu et al., 2024). This disparity requires a methodical selection process to ensure that the data contributing to model training are relevant and of high quality.

While addressing these challenges, this paper specifically focuses on two fundamental objectives: maximizing global model performance and minimizing energy consumption in parallel. Maximizing global performance ensures that the federated model performs well across diverse datasets, which is essential for the practical applicability of FL (Mora et al., 2024). Minimizing total energy consumption from all clients yields substantial benefits (Tang et al., 2024). Beyond just extending device battery life, choosing clients that consume less energy to participate in FL also reduce the overall energy consumption, thereby reducing the operational costs and enhancing the sustainability of network infrastructures.

In this paper, we present a novel context-aware client selection mechanism that employs a multi-objective optimization framework, significantly enhancing client selection beyond current state-of-the-art methods. The key contributions are summarized as follows:

**Novel Contextual-Aware Multi-Objective Multi-armed Bandit Client Selection Framework:** We introduce Pareto Contextual Zooming for Federated Learning (PCZFL), a novel approach adapted from foundational concepts in the Multi-Armed Bandit (MAB) literature (Turgay et al., 2018). Our proposed framework is specifically designed to optimize client selection by balancing multiple dimensions, such as accuracy and energy efficiency, and with theoretical justifications. In comparative evaluations, PCZFL demonstrates superior performance over existing state-of-the-art methods across these metrics.

**Contextual Information Storage and Utilization:** Our framework continuously monitors and adapts to the dynamic contexts of clients by organizing them into structured groups based on their contextual similarities. This approach not only preserves historical context but also facilitates the tracking of evolving patterns in client performance.

**Extensive Experimental Studies:** We conducted extensive experiments to evaluate both accuracy and energy outcomes throughout the entire training process under varying levels of uncertainty coefficient. The experimental results clearly demonstrate that our method significantly surpasses existing state-of-the-art approaches, achieving superior performance in both accuracy and energy efficiency.

## 2 Related Work

Effective client selection in FL enhances model convergence, minimizes communication costs, and ensures diverse data representation. Cho et al. (2022) propose the Power-of-Choice strategy, using local loss as a contribution score to prioritize clients, enhancing FL speed and accuracy, yet the accuracy of local loss estimates and their sole focus could limit broader client contributions. Lastly, Balakrishnan et al. (2022) introduce a strategy to select a diverse subset of clients by maximizing a submodular facility location function over gradient space, improving convergence, fairness, and efficiency. However, this approach's pursuit of diversity could occasionally reduce efficiency and the computational demands for evaluating submodular functions and updating gradients pose challenges in large-scale systems.

While diverse strategies have been explored for optimizing client selection in FL, another significant line of research involves the use of Multi-Armed Bandit (MAB) frameworks, which provide a dynamic mechanism to address the complexities of client variability and the optimization of multiple performance metrics. Lai et al. (2021) prioritize clients with the most useful data and quickest training capabilities to improve time-to-accuracy performance significantly and enhance final model accuracy. However, Oort does not account for the contextual information of each client, which could lead to slower global model convergence and adaptability issues in dynamic FL environments. Similarly, Ami et al. (2023) employ a bandit-based approach to client selection, focusing on minimizing latency and improving model generalization. By using a weighted sum to combine these objectives within the Upper Confidence Bound framework, BSFL effectively balances the trade-off between quick training iterations and the quality of the global model. While this method integrates multiple performance aspects, its adaptability to changing network conditions and client states could be enhanced by incorporating contextual data that reflects real-time performance variations. Taking a different approach, Jung et al. (2022) introduce a Pareto optimality-based approach to handle the trade-offs between

resource efficiency and model convergence in mobile networks. By prioritizing clients that provide the best compromise between resource consumption and training effectiveness, the approach aims to enhance both the speed and the quality of the learning process. However, while this method efficiently balances between competing objectives, it does not incorporate real-time contextual data, which could further optimize client selection based on dynamic network conditions. Ma et al. (2024) introduce a context-aware algorithm that leverages a combinatorial contextual neural bandit framework that emphasizes the enhanced extraction of contextual information from clients, evaluated against a universally standardized dataset, in order to produce a more insightful contextual representation tailored for federated settings. Nunes et al. (2024) address the dual challenges of time efficiency and energy conservation in client selection processes by introducing two innovative algorithms: Minimal Makespan and Energy Consumption FL Schedule (MEC) and Minimal Energy Consumption and Makespan FL Schedule under Time Constraint (ECMTC). While it adds an extra step to calculate the optimal number of clients for selection, it also introduces significant computational overhead and it may only be limited to the aspect of time and energy and not adaptive to other potential dimensions of factors.

## 3 Background & Problem Formulation

Our framework is based on an FL architecture that involves $N$ clients, where each client $i \in \{1, 2, \ldots, N\}$ has a unique dataset $D_i$ , using which the FL system train individual models and synthesizes a comprehensive global model suitable for tasks. In each iteration $t = 1, 2, \ldots, T$ of the FL process, the server sends the global model parameters $w_t$ to a subset of clients selected for training. These clients use their specific dataset $D_i$ to refine the model using methods such as stochastic gradient descent (SGD), resulting in updated model parameters $w_{t+1}^{(i)}$ which are then sent back to the server and aggregated to form the revised global model $w_{t+1}$.

In this work, we formulate the problem of client selection as a contextual MAB problem, where we treat the server as the gambler, aiming to maximize the utility, while the selection of the clients becomes choosing an arm from the slot machines, and each arm offers a distinct reward (or utility) upon being "pulled". However, unlike traditional MAB problems, our scenario involves multiple objectives that need simultaneous optimization. Specifically, the server aims to improve the accuracy of the global model while minimizing the total accumulated energy consumption throughout the training process. Each client's potential contribution to these objectives is only partially known at the start and becomes more apparent through ongoing interactions. By employing a multi-objective MAB-based approach (Turgay et al., 2018), we effectively navigate the decision-making process in this partially known environment, enabling the server to make more informed and strategic choices over time.

At the beginning of round $t$, the system observes a context $X_t$ from a context space $\mathcal{X} = \mathbb{R}^{N \times d_r}$, where $d_r$ is the number of objectives we care about. For instance, $d_r = 2$ when we have accuracy and energy efficiency as our objectives. The system then selects an arm/client $y_t$ from the set of one-hot vectors $\mathcal{Y} = \{y : y \in \{0, 1\}^N, \sum_{i=1}^N y_i = 1\}$, indicating which client would be picked for training. It is preferable for the server to pick multiple clients for stable training in each round, so our algorithm will do so once it identifies the Pareto front (discussed later). Upon selecting the clients, the system receives a $d_r$-dimensional reward vector $r_t = [r_t^1, \ldots, r_t^{d_r}]$, where $r_t^j$ represents the reward from the $j$-th objective in round $t$. Define $\mu_y(X) \in \mathbb{R}^{d_r}$ to be the vector of the underlying expected rewards when $y$ is selected given context $X$. We assume that the reward vector $r_t$ obtained from the selected $y_t$ satisfies $r_t = \mu_{y_t}(X_t) + \kappa_t$, with $\kappa_t$ being a $d_r$-dimensional noise vector. This noise process for each objective is assumed to have a marginal distribution that is conditionally 1-sub-Gaussian: $\mathbb{E}[e^{\lambda \kappa_t^j} \mid y_{1:t}, X_{1:t}, \kappa_{1:t-1}] \leq \exp(\lambda^2/2)$ for any $\lambda \in \mathbb{R}$, where $b_{1:t} = (b_1, \ldots, b_t)$.

Finally, we assume that the expected reward is Lipschitz w.r.t. the context vector:

**Assumption 1.** *Let* $j \in \{1, \ldots, d_r\}$, $X, X' \in \mathcal{X}$ *and* $y, y' \in \mathcal{Y}$. *We assume that, for some* $0 < C < \infty$ :

$$\left| \mu_y^j(X) - \mu_{y'}^j(X') \right| \leq \|X^\top y - X'^\top y')\|_2 \leq C \tag{1}$$

This assumption allows us to upper bound the reward distance using the Euclidean distance in $\mathbb{R}^{d_r}$, which is critical to obtain the regret bound theoretically. In the following, we refer to $(\mathbb{R}^{d_r}, \|\cdot\|_2)$ the *similarity space*.

In FL, where multiple objectives such as model accuracy and energy efficiency must be balanced across decentralized devices, Pareto optimality provides a fundamental framework for client selection, indicating that one cannot improve its performance without compromising another.

**Definition 1** (Pareto Optimality). *1. y is weakly dominated by $y'$ under context $X$, denoted by $\mu_y(X) \preceq \mu_{y'}(X)$, if $\mu_v^j(X) \leq \mu_{v'}^j(X), \forall j \in \{1, \ldots, d_r\}$.*

*2. y is dominated by $y'$ under context $X$, denoted by $\mu_y(X) \prec \mu_{y'}(X)$, if it is weakly dominated and $\exists j \in \{1, \ldots, d_r\}$ for which $\mu_r^j(X) < \mu_{y'}^j(X)$.*

*3. y and $y'$ are incomparable under context $X$, denoted by $\mu_y(X) \parallel \mu_{y'}(X)$, if neither dominates the other.*

*4. y is Pareto optimal under context $X$ if no other $y'$ dominates it under the same context. The set of Pareto optimal y for a specific context $X$ forms the* Pareto front.

Identifying the Pareto front is crucial for selecting effective clients and maintaining training stability, but it is also challenging in high-dimensional spaces with multiple objectives. To address this, we rely on the contextual zooming algorithm (Slivkins, 2011; Turgay et al., 2018), which adaptively zooms in on critical regions in the similarity space. It involves grouping selections based on their contextual similarities, creating what are referred to as "balls". By adaptively pooling observations from clients who share similar contexts, we effectively mitigate the issues associated with sparse data and reduce the computational burden by decreasing the number of direct comparisons required, as Pareto optimality is applied to these grouped entities rather than individuals.

**Definition 2** (Domain of a Ball). *Given a set of balls $\mathcal{B}$ in $\mathbb{R}^{d_r}$, the domain of a ball is defined as:*

$$dom_{\mathcal{B}}(B) := B \setminus \left( \bigcup_{B' \in \mathcal{B}: r(B') < r(B)} B' \right) \tag{2}$$

*where $r(B)$ indicates the radius of the ball $B$, and the union encompasses all balls $B'$ within $\mathcal{B}$ that have a smaller radius than $B$.*

A numerical toy example is presented in Figures 1 and 2, which depict the balls in the similarity space defined by local model accuracy and energy score (in percentage), the latter being $1 -$ energy consumption. This transformation was applied to ensure that higher values on both axes represent better results, aligning the direction of improvement across dimensions. The domain of each ball is visually delineated with dashed lines within its boundary. Note that the domain of Ball a excludes Ball d, which will be important to determine which domain a client falls into later.

**Energy consumption computation**. In managing the operational efficiency of FL systems, it is critical to account for energy consumption. Following the existing methodologies (Zhou et al., 2023; Zheng et al., 2021), we compute the energy consumption for each client $i$, with the assumption that the energy does not change throughout the FL training.

## 4 Pareto Contextual Zooming for FL

We introduce a new client selection mechanism for FL, termed PCZFL, which leverages the concept of Pareto contextual zooming (Turgay et al., 2018). Our algorithm aims to identify clients proximal to the Pareto front by adaptively partitioning similarity space. In the subsequent sections, we will begin with a detailed conceptualization of PCZFL, followed by a numerical example that illustrates the procedure.

### 4.1 Context Extraction of PCZFL

To enhance the context extraction for each client, we use a validation dataset maintained on the server side. This dataset is utilized to evaluate the clients' models, akin to conducting an "interview" for each client.

This approach builds upon an existing work where the concept of an interview dataset was used to extract context effectively (Ma et al., 2024). Specifically, the server assesses the models against this standardized dataset, and the resulting accuracy provides critical contextual information for each client. It is important to note that the clients do not have access to this dataset, ensuring that the server's evaluation remains objective and unbiased.

In our framework, the context for each client $i$ at round $t$ is represented by two key metrics: the accuracy $A_{t,i}$ of the client's local model on the interview dataset, and the total energy consumption $E_{t,i}$:

$$x_{t,i} = [A_{t,i}, E_{t,i}] \tag{3}$$

For clients selected in the current round, their context vectors are freshly computed using Equation 3. Clients not selected retain their context from previous rounds, specifically from the most recent round in which they participated. These context vectors then inform the subsequent steps of our algorithm. The complete context information at round $t$ is then $X_t$, a matrix stacking $x_{t,i}$.

## 4.2 Algorithm Description of PCZFL

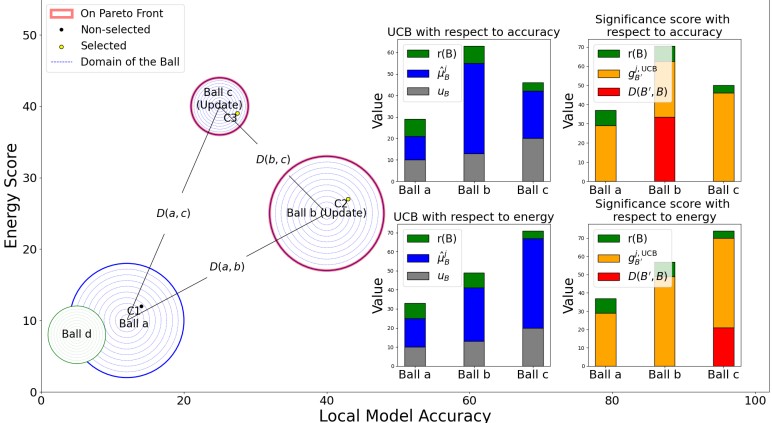

Figure 1: Selection process at time $t$. The left subfigure illustrates the current ball structure and client positions. Each ball represents a region of the similarity space. Dashed blue lines indicate the balls' domain as in Definition 2. Note that Ball `d` is not included in the domain of Ball `a`, since Definition 2 specifies that a ball's domain excludes any other ball with a smaller radius that overlaps with its region. At this time step, three balls (`a`, `b`, and `c`) are *relevant* because they contain at least one client context (points `C1`–`C3`). Ball `d` is *irrelevant* because no client falls into its domain. The middle subfigure decomposes the UCB calculation for each ball according to Equation 4, where blue corresponds to the sample mean reward, gray to the sample uncertainty term, and green to the radius. The right subfigure shows the resulting significance scores from Equation 5. Every ball $B$ will need to identify a related ball $B'$ with the minimum combining score of $g_{B'}^{j,\text{UCB}}$ and $D(B',B)$.

Algorithm 1 delineates the procedural details of PCZFL. This algorithm adaptively partitions the similarity space in a non-uniform manner based on the clients chosen in each round of FL. The partitioned space comprises a set of active balls, denoted by $\mathcal{B}$, which encapsulates the contextual information and rewards observed in previous rounds. Importantly, the composition of $\mathcal{B}$ may vary with each FL round. Each active ball $B \in \mathcal{B}$ is characterized by a radius $r(B)$, a center and a domain $\text{dom}_{\mathcal{B}}(B)$ as defined in Definition 2. The dynamic nature of these balls allows for a flexible and responsive adaptation to the evolving landscape of client performance and relevance. An illustration of some balls at time $t$ and $t+1$ can be found in Figure 1 and 2. Intuitively speaking, these balls group similar clients so that one can infer future contexts from historical contexts.

Initially, PCZFL is configured with the total number of training epochs $T$. An initial ball $B_0$ is created to cover the entire similarity space, centered at an arbitrarily chosen point. At the beginning of each FL epoch,

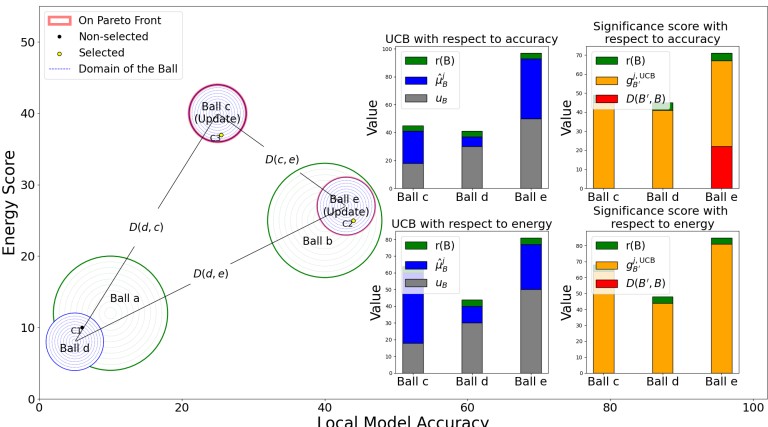

Figure 2: Selection process at time $t+1$. After observing new contextual points, PCZFL refines the ball structure by applying the zooming step. A new ball (Ball `e`) is generated around the previous client `C2` because the uncertainty of Ball `b` became sufficiently small, triggering subdivision. As a result, the set of relevant balls has shifted: Balls `c`, `d`, and the newly created Ball `e` are now relevant at this time step, while Ball `b` becomes irrelevant. The updated Pareto front (red boundaries) is recalculated, and the selected clients for this round are determined accordingly.

PCZFL observes the context $X_t$ and identifies the set of relevant balls containing this context, denoted by

$$\hat{\mathcal{R}}(X_t) := \{B \in \mathcal{B} : X_t^\top y \in \mathrm{dom}_{\mathcal{B}}(B) \text{ for some } y \in \mathcal{Y}\}.$$

As depicted in Figure 1, Balls `a`, `b`, and `c` are relevant since there is at least a client's context (`C1`, `C2` or `C3`) falls in it, while Ball `d` is irrelevant in this round since there is no client falls into its domain. It is important to mention that the composition of relevant balls may shift as the FL rounds progress due to the dynamic nature of each round's context. Hence, as illustrated in Figure 2, by time $t+1$, Balls `c`, `e`, and `d` become the new set of relevant balls.

In a dynamic FL environment characterized by heterogeneous and evolving client attributes, employing the principle of "optimism in the face of uncertainty" is particularly advantageous (Guo et al., 2024). This principle, akin to assuming the most favorable conditions in MAB problems, inflates the reward estimates, thereby creating an upper confidence bound (UCB) for the expected reward. In the PCZFL framework, we leverage the UCB as a crucial component to assess and identify the most promising "balls" within the similarity space. Additionally, it is essential to compare and adjust the UCB of each ball against others in the similarity space to ensure the system avoids selecting locally optimal solutions that may not be globally efficient.

The UCB for each ball $B$ with respective to the $j$-th objective is calculated as follows:

$$g_B^{j,\mathrm{UCB}} := \hat{\mu}_B^j + u_B + r(B), j \in \{1, ..., d_r\} \quad \text{where} \tag{4}$$

- $\hat{\mu}_B^j$ represents the sample mean reward for ball $B$.
- $u_B = \sqrt{2A_B/N_B}$ denotes the sample uncertainty where $A_B$ is $(1 + 2\log(2\sqrt{2}d_r T^{\frac{3}{2}}/\delta))$ and $N_B$ is the number of times ball $B$ has been selected. This term quantifies the uncertainty about the mean estimate with $\delta \in (0, 1)$ being the confidence level. In theory, $A_B$ is specifically chosen to achieve a high probability regret bound but in the experiment, we treat it as a hyper-parameter. The uncertainly of a ball decreases as we observe more reward for it.
- $r(B)$ represents the radius of ball $B$, which encapsulates the contextual uncertainty due to the variability of contexts within ball $B$. This radius remains constant for each ball throughout the FL rounds.

The UCB scores are shown in the middle subplots of Figure 1, where each color corresponds to a term in Equation 4. We can see that Ball `c` has a smaller radius (green) than Balls `a` and `b`. The sample mean reward (blue) and uncertainty (gray) are calculated according to the historical rewards and counts.

Similar to Turgay et al. (2018), PCZFL determines the Pareto front based on a proxy, which we call the *significance score* for each ball $B$ with respective to the $j$-th objective computed as:

$$g_B^j := r(B) + \min_{B' \in \mathcal{B}} \left( g_{B'}^{j,\text{UCB}} + D(B', B) \right), j \in \{1, ..., d_r\} \tag{5}$$

where $D(B', B)$ represents the distance between the centers of $B'$, a neighbour ball of $B$, and $B$ in the similarity space. This equation not only considers the inherent qualities and uncertainties of ball $B$ but also relates these qualities to those of other balls, ensuring a comparison that promotes the selection of balls which are non-dominated on the Pareto front. Theoretically, this significance score offers a way to relate to the expected reward, leading to a meaningful regret bound (Turgay et al., 2018).

Upon computing the significance scores for the set of relevant balls, PCZFL delineates the Pareto front within $\hat{R}(X_t)$ by using $g_B := [g_B^1, ..., g_B^{d_r}]$ as a proxy for expected rewards, defined as follows:

$$\hat{A}_t^* := \{B \in \hat{\mathcal{R}}(X_t) : g_B \nprec g_{B'}, \forall B' \in \hat{\mathcal{R}}(X_t)\}$$

$\hat{A}_t^*$ captures the balls that are not dominated by any other balls. To stabilize training, instead of selecting one single client, here we choose all clients covered by $\hat{A}_t^*$ to conduct local training. The local models from these selected clients are then aggregated to form the global model.

These calculations are illustrated on the right subplots in Figure 1, where each color represents a term in Equation 5. It is important to note that the significance score is derived from a minimization. Consequently, the distance $D(B', B)$ may be zero if $B'$ is determined by the argmin as being equal to $B$. In the left plot, balls outlined in red are determined to be on the Pareto front, while the contextual points highlighted in yellow indicate that the corresponding clients (`C2` and `C3`) have been selected for local model upload in FL aggregation.

After observing the rewards from the selected clients, the algorithm proceeds to update estimates and counts for all the balls from the Pareto front corresponding to the selected clients. Finally, PCZFL adopts a contextual zooming strategy for each chosen ball $\hat{B} \in \mathcal{B}$, which dynamically partitions the context space based on historical context arrivals and corresponding rewards. Specifically, if the sample uncertainty for a selected ball becomes lower than the ball's radius in a given FL round, a new ball $B'$ is created with half the radius of $B$ and a center corresponding to the contextual point of ball $B$. Intuitively, since the uncertainty becomes small enough, one may want to "zoom in" and have a more refined view of this specific area of the similarity space. Accordingly, the domain space that covers of all the balls are updated. In addition, the sample mean $\hat{\mu}_{\hat{B}}^j$ and the count of occurrences for $\hat{B} \in \hat{A}_t^*$ are also updated. It is important to highlight that the hyper-parameter $A_B$ fulfills dual roles. Firstly, as demonstrated in Equation 4, $A_B$ balances between the sample mean $\hat{\mu}_B$ and the sample uncertainty $u_B$. Secondly, $A_B$ also dictates the rate at which new balls are generated, as specified in line 12 of Algorithm 1. As in the running example in Figure 2, Ball `e` emerges as a new ball and centered around the contextual point of client `C2` from the previous round (as in Figure 1). This generation occurs because the uncertainty of Ball `b` is low enough to trigger ball generation. Furthermore, Ball `e` supersedes Ball `b` at time $t+1$ for the selection of relevant balls because `C2`'s new position falls within the domain of Ball `e`.

### 4.3 Theoretical Analysis

**Regret Analysis**. Since PCZFL extends the PCZ algorithm (Turgay et al., 2018) to the FL client selection setting, we follow the same regret analysis framework as presented in Turgay et al. (2018). The abstract distance metric $D$ in Assumption 1 of Turgay et al. (2018) is now replaced by the Euclidean distance due to our meticulously designed context and arm domains $\mathcal{X}, \mathcal{Y}$ (i.e., $D\left((X, y), (X', y')\right) = \|X^\top y - X'^\top y'\|_2$). These design choices ensure that the Lipschitz condition remains valid throughout the proofs of Turgay et al. (2018) up to constant factors. As a result, our algorithm achieves the sub-lienar regret bound given

---

**Algorithm 1** PCZFL

---

1: **Input:** Number of epochs $T$, initial model $w_0$
2: **Output:** Final model $w_T$
3: **Initialize:** A collection set $\mathcal{B} = \{B_0\}$ of "active balls" where $B_0$ covers the whole similarity space, counters $N_B = 0$ and estimates $\hat{\mu}_B^j = 0, \forall B \in \mathcal{B}, \forall j \in \{1, \ldots, d_r\}$
4: **for** each epoch $1 \leq t \leq T$ **do**
5:      Observe $X_t$
6:      $\hat{\mathcal{R}}(X_t) \leftarrow \{B \in \mathcal{B} : X_t^\top y \in \mathrm{dom}_{\mathcal{B}}(B) \text{ for some } y \in \mathcal{Y}\}$
7:      $\hat{A}_t^* \leftarrow \{B \in \hat{\mathcal{R}}(X_t) : g_B \nprec g_{B'}, \forall B' \in \hat{\mathcal{R}}(X_t)\}$
8:      Select clients $\{i : x_{t,i} \in \bigcup_{B \in \hat{A}_t^*} \mathrm{dom}_{\mathcal{B}}(B)\}$ to conduct local training
9:      Aggregate the models to form $w_t$
10:      Observe the rewards $r_t^j, \forall j \in \{1, \ldots, d_r\}$
11:      **for** chosen ball $\hat{B}$ **do**
12:          **if** $u_{\hat{B}} \leq r(\hat{B})$ **then**
13:              Create a new ball $B'$ whose center is its corresponding contextual point and radius is $r(B') = r(\hat{B})/2$
14:              $\mathcal{B} \leftarrow \mathcal{B} \cup B', \hat{\mu}_{B'}^j = \mathrm{Center}(B'), \forall j, N_{B'} = 0$
15:              Update $\mathrm{dom}_t(\mathcal{B})$
16:          **end if**
17:          $\hat{\mu}_{\hat{B}}^j \leftarrow (\hat{\mu}_{\hat{B}}^j N_{\hat{B}} + r_t^j)/(N_{\hat{B}} + 1), \forall j$
18:          $N_{\hat{B}} \leftarrow N_{\hat{B}} + 1$
19:      **end for**
20: **end for**

---

by Theorem 1 and Corollary 1 of Turgay et al. (2018), which means that with high probability, the Pareto regret is $\widetilde{O}\big(T^{(1+d_z)/(2+d_z)}\big)$, where $d_z$ is the Pareto zooming dimension.

**Time Complexity Analysis**. The time complexity of the PCZFL algorithm in each round stems from the computation cost incurred by each client, as well as the overhead associated with the client selection procedure. Assuming each basic operation requires one time unit, the complexity of each individual component is detailed as follows:

1. **Obtaining Contextual Information and True Reward Calculation:** Each client derives its accuracy context by assessing its locally trained model's per-class losses on the interview dataset. Let $N_{\mathrm{sel}}$ denote the number of selected clients per FL round. This step incurs a time complexity of $O\big(N_{\mathrm{sel}} \times (M + E)\big)$, where $M$ represents the time units spent evaluating the model's loss and $E$ represents the time units spent calculating energy consumption. The same procedure is then used to compute the true reward.

2. **Pareto Front Calculation:** The computational complexity of this operation increases as the number of balls increases. According to Kung et al. (1975), this complexity can be expressed as: $b \times \log b$ if $d_r \leq 3$ or $b \times (\log b)^{d_r - 2}$ if $d_r > 3$, where $b$ represents the number of balls considered, and $d_r$ denotes the number of dimensions of the objective. Note that since at most one sub-ball can be generated per round, the total number of balls $b$ is bounded by the number of rounds $T$.

3. **Client Selection:** All clients from balls on the Pareto front are included directly in the federated averaging process. The selection process does not add any computational complexity beyond identifying the number of balls on the Pareto front, which is $\widetilde{O}(b)$ assuming that no additional processing is performed on the client list.

Given that the dimension $d_r$ in the work is 2, the total time complexity in one round is

$$O(N_{\mathrm{sel}} * (M + E) + T \log T).$$

# 5 Empirical Evaluation

In this section, we present our experimental results for PCZFL, comparing it against several baseline approaches. To ensure consistent comparisons across different baselines, the same seeds was used for all experiments. We averaged the results over three runs to improve the reliability of our findings. We evaluate our methods using global accuracy and total energy consumption, comparing them to the baselines. For these metrics, higher global accuracy and lower energy consumption are considered better. During the early stages of the FL process, it is critical to have sufficient exploration to know enough about the clients. This initial phase guarantees that the global model incorporates a diverse set of local updates, reflecting a broad spectrum of the data distribution. Drawing on key findings from foundational research (Yan et al., 2022), we let all clients to participate in the first 20 epochs, recognizing this period as vital for establishing a robust model and is consistently applied across all baselines.

## 5.1 Model Setup

We use PyTorch to implement the local models for the clients. The local models are built as convolutional neural networks (CNNs) for image data. The detailed local model architecture closely follows the design specified in Ma et al. (2024). Our experiments were conducted on multiple virtual machines, each configured with high-performance GPUs, including Nvidia Titan V, RTX 3060, and RTX 4090 models, all operating under Ubuntu 22.04.

## 5.2 Data Preparation

To evaluate the effectiveness and robustness of our proposed PCZFL framework, we conduct experiments on two widely used image classification benchmarks: CIFAR-10 (Krizhevsky, 2009) and Fashion-MNIST (FMNIST) (Xiao et al., 2017). CIFAR-10 comprises 60,000 32x32 color images, evenly distributed across 10 classes, each of which containing 6,000 images. We derive the interview dataset by extracting 20% from the test dataset, which comprises 10,000 samples, while the training dataset includes 50,000 samples. The remaining 80% of the test dataset, not used for the interview, serves as the test data. FMNIST is a grayscale image dataset comprising 70,000 28×28 images across 10 clothing categories. It is divided into 60,000 training and 10,000 test images. To align with our methodology, we again extract 20% of the test set (2,000 samples) as the interview dataset, reserving the remaining 8,000 samples for evaluation. Compared to CIFAR-10, FMNIST is less complex in terms of image resolution and content but introduces different challenges due to class imbalance under heterogeneous client distributions.

## 5.3 Baselines for Comparison

In our study, we benchmark our method against five distinct approaches: AFL (Goetz et al., 2019), DivFL (Balakrishnan et al., 2022), Power-of-choice (Pow-d) (Cho et al., 2022), LSH and NCCB (Pan et al., 2023). AFL improves the selection process by introducing a value function that evaluates the utility of each client's data in relation to the current model. This assessment adjusts the probability of a client's inclusion in training, aiming to reduce the number of iterations required to achieve predefined accuracy levels by 20-70% without compromising the model's performance. DivFL enhances the diversity of client selection reducing the variability introduced by selecting subsets, which could slow the learning trajectory. It employs submodular maximization of a facility location function defined on the gradient space to achieve this, thereby promoting faster convergence and fairness among participants. Pow-d strategy selects clients that show higher local losses, which can speed up the convergence on errors. LSH serves as a simplified version of the NCCB approach. It replicates NCCB's method for extracting context by applying Simhash to local training datasets but omits the clustering algorithm, which is a key component of the full NCCB method. We then extend the LSH approach to encompass the full NCCB method by implementing k-means clustering with Hamming distance as the metric to evaluate hash values. To adapt these baselines, we have modified their original utility scores by replacing them with a weighted average of validated accuracy and energy consumption. These revised utility scores are then utilized to proceed with client selection.

### 5.4 Energy Setting

In our setting, it is assumed that each client has a random but fixed level of energy consumption per round throughout the training process. This setting provides a controlled environment by removing the fluctuations in energy consumption on each client during each training period, and allows us to isolate the effects of algorithmic changes without being confounded with fluctuating energy usage.

### 5.5 Experiments and Result Analysis

In our experiment, we employ a Dirichlet-mapped data distribution to partition client data, renowned for its effective representation of non-i.i.d. datasets. This method is also ideal for simulating practical environments in real-world heterogeneous networks (Mayhoub & Shami, 2023). Specifically, we set the Dirichlet concentration parameter to $\alpha = 0.1$, which results in highly skewed local class distributions across clients and thus simulates strong heterogeneity in the federation. For evaluation, we focus on two key metrics: (i) *global accuracy*, defined as the test accuracy of the aggregated global model, and (ii) *accumulated energy consumption*, calculated as the sum of the per-round energy consumption of all selected clients across the entire training process, where lower values indicate improved energy efficiency.

#### 5.5.1 Behavior of PCZFL on the Similarity Space

In our experiment setup, we use a total of 30 clients. It is important to note that in PCZFL, the number of clients selected per FL epoch is dynamic and determined by Pareto optimality, resulting in variability in the selection across different rounds. To align this aspect with all other baseline methods, we set the number of selected clients to match the average selection rate of PCZFL throughout the training period, which is approximately 5 clients per FL epoch.

Figure 3 visualizes the evolution of our algorithm in the similarity space at epochs 0, 50 and 200. Each green point represents a client with two metrics: local model accuracy (evaluated in a held-out validation dataset) and an energy score, defined as $(1-$ the normalized energy consumption) in the current FL round. This formulation ensures that both axes follow a consistent interpretation – higher values reflect better performance. At epoch 0, prior to any training, all client models are randomly initialized, resulting in validation accuracies clustered around the untrained baseline of approximately 0.1 accuracy. Meanwhile, to avoid skewed distributions in energy scores, we uniformly sample the energy score from $[0.5, 1]$, producing a compact cluster of points in the top-left region of the space. The only related ball (blue circle) in the beginning of training is the uninformed one, including all possible clients in the entire space. As training progresses to epoch 50, there is a noticeable spread in the client points along the accuracy axis. This spread indicates a divergence in model performances as individual clients learn from their distinct data distributions. Smaller, more defined clusters begin to emerge (highlighted by blue balls), encompassing localized groups of similar models. The red balls identify clients on the Pareto front, those achieve a good trade-off between accuracy and energy efficiency. By epoch 200, these clusters become more distinct and compact, some containing only a single client, indicating highly specialized models. The Pareto front (red circles) now encompasses diverse trade-offs, reflecting the system's ability to discover multiple effective strategies for balancing model performance and energy usage.

#### 5.5.2 Comparative Analysis against Baselines

**CIFAR-10 Results**

Figure 4 demonstrates the global accuracy comparison of PCZFL against the baselines on CIFAR-10. It is evident that PCZFL achieves superior global accuracy relative to the other approaches. Conversely, LSH consistently shows lower global accuracy, not only in comparison to PCZFL but also relative to other baseline models throughout the training epochs. This lower performance may be attributed to LSH's inability to capture sufficiently enriched context information within this experimental setting. However, its successor, NCCB, demonstrates higher global accuracy with the implementation of its inherited clustering algorithm, indicating a notable improvement. It is important to mention that the significant drop observed around epoch 20 is linked to the initial period setting, which mandates the participation of all clients. The sudden

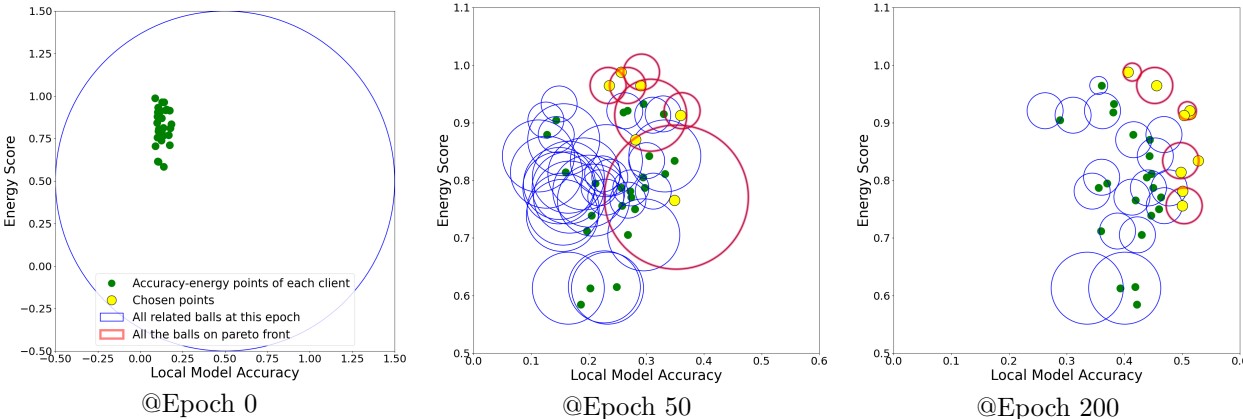

@Epoch 0          @Epoch 50          @Epoch 200

Figure 3: Evolution of Balls at Different Epochs

reduction in the number of participating clients from 30 to 5 likely causes a drop in global accuracy, as there is less data contributing to the overall training process. Quantitatively, PCZFL achieves a **4.7% higher global accuracy** ($\frac{0.5389-0.5145}{0.5145} \times 100 = 4.7\%$) compared to Pow-d, the second-best baseline, and improves upon NCCB by **9.5%** in global accuracy ($\frac{0.5389-0.4921}{0.4921} \times 100 = 9.5\%$).

Figure 5 presents a comparative analysis of accumulated energy consumption over the training rounds on CIFAR-10. As depicted, PCZFL not only outperforms all baseline models in terms of global accuracy but also exhibits significantly lower energy consumption. NCCB ranks second, offering a commendable balance between global accuracy and energy efficiency. The remaining four baselines show similar trends, with Pow-d recording the highest accumulated energy consumption. Overall, PCZFL reduces the total energy cost by **37.3%** compared to Pow-d ($\frac{332.23-208.15}{332.23} \times 100 = 37.3\%$) and by **30.9%** compared to NCCB ($\frac{301.54-208.15}{301.54} \times 100 = 30.9\%$).

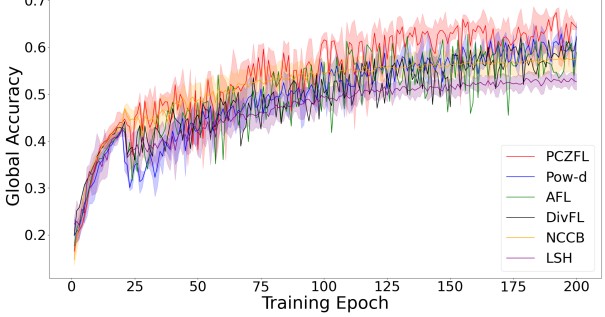

Figure 4: Global Accuracy Comparison on CIFAR-10. The shaded regions represent the standard deviation across three independent runs.

Figure 5: Accumulated Energy Consumption on CIFAR-10. The shaded regions represent the standard deviation across three independent runs.

**FMNIST Results**

To further validate the robustness of PCZFL, we repeat the experiments on FMNIST. Figure 6 illustrates the global accuracy comparison. Since FMNIST is less complex than CIFAR-10, we reduced the mandatory participation period from 20 rounds to 5 rounds to prevent the model from being overtrained at the starting point. As a result, a noticeable accuracy drop occurs immediately after epoch 5, mirroring the CIFAR-10 trend but at an earlier stage. Furthermore, due to the dataset's lower complexity, global accuracy converges to a stable plateau much earlier than in CIFAR-10, with fluctuations becoming minimal after the mid-stage of training. Quantitatively, PCZFL achieves a **3.2% improvement in global accuracy** ($\frac{0.919-0.8905}{0.8905} \times 100 = 3.2\%$) over Pow-d and a **3.7% improvement** over NCCB ($\frac{0.919-0.8863}{0.8863} \times 100 = 3.7\%$), confirming its consistent advantage across benchmarks.

Figure 7 reports the accumulated energy consumption on FMNIST. Consistent with the CIFAR-10 results, PCZFL delivers the most energy-efficient training process, significantly outperforming Pow-d and LSH, which consume far more energy. NCCB again serves as the closest competitor in terms of efficiency but still trails behind PCZFL in both accuracy and energy savings. Overall, PCZFL reduces the total energy cost by **21.4%** relative to Pow-d ($\frac{364.87-286.71}{364.87} \times 100 = 21.4\%$) and by **30.8%** relative to NCCB ($\frac{414.16-286.71}{414.16} \times 100 = 30.8\%$).

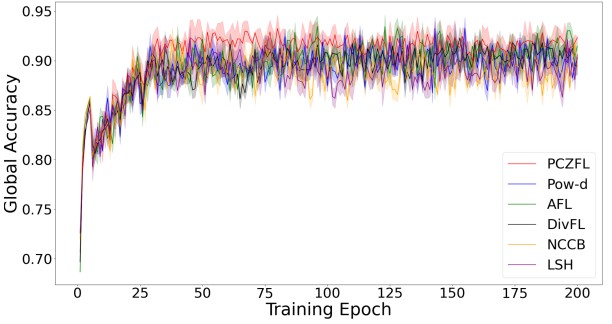

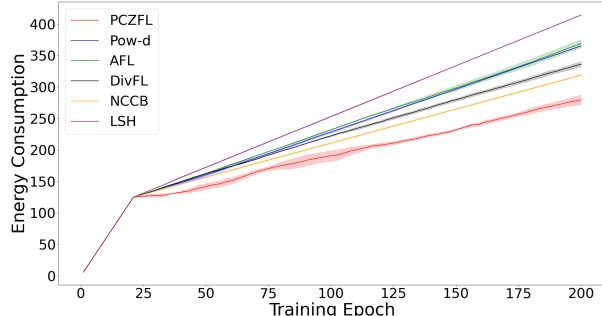

Figure 6: Global Accuracy Comparison on FMNIST. The shaded regions represent the standard deviation across three independent runs.

Figure 7: Accumulated Energy Consumption on FM-NIST. The shaded regions represent the standard deviation across three independent runs.

### 5.5.3 Ablation Study on the Uncertainty Coefficient

The coefficient of sample uncertainty, $A_B$, is a critical hyperparameter in PCZFL, significantly influencing the exploration aspect of our client selection mechanism and the rate at which new balls are formed as training progresses (see line 12 of Algorithm 1).

Figures 8 and 9 showcase the global accuracy and accumulated energy consumption for various uncertainty coefficients on CIFAR-10. From Figure 8, it's evident that the variance in global accuracy among different coefficients is minimal, though smaller coefficients seem to provide a marginal benefit. This slight advantage indicates that lower uncertainty coefficients contribute positively, albeit subtly, to model performance. Conversely, Figure 9 reveals a more pronounced trend in energy consumption. As the uncertainty coefficient increases, not only does the energy consumption escalate, but also the variability in accumulated energy usage across distinct training runs becomes more pronounced. The rationale for these observations is twofold: 1. A smaller uncertainty coefficient reduces the influence of sample uncertainty in the UCB score computation, favoring the sample mean and contextual uncertainty. This coefficient choice aligns with our experimental setup, where accuracy and energy conditions remain stable across FL epochs; 2. Lower coefficients also imply a quicker rate of creating new balls in the similarity space, facilitating faster convergence. This is particularly advantageous in our experimental setting, where the dynamics within the accuracy and energy landscape are relatively stable, thus not necessitating frequent adjustments based on dramatic shifts in the context.

## 6   Conclusion

Our PCZFL approach represents a significant advancement in addressing the dual challenges of efficiency and effectiveness with FL. By striking a balance between model accuracy and energy consumption, PCZFL not only aligns with the practical demands of modern distributed computing environments but also promotes sustainability in ML deployments.

For future work, we plan to revise our approach by considering a more flexible arm set $\mathcal{Y}$ such as the probability simplex so that one can directly map an arm to the combining coefficients of the clients. Furthermore, we aim to investigate the adaptability of PCZFL in scenarios involving volatile clients (Shi et al., 2022), whose participation is irregular and unpredictable due to factors like device availability, network connectivity, or energy constraints. Understanding how PCZFL can maintain its robustness and performance in the face of such volatility will be crucial to enhance its practicality and reliability in real-world federated networks.

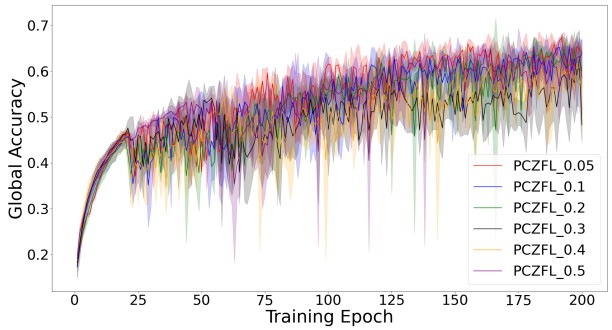
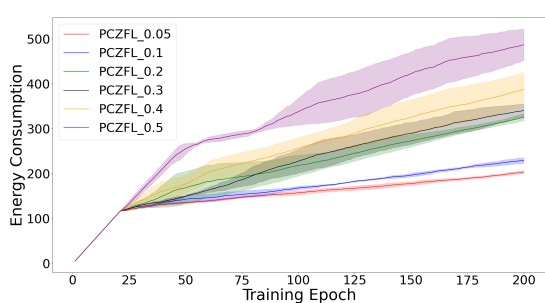

Figure 8: Global Accuracy for different coefficient uncertainties on CIFAR-10. The shaded regions represent the standard deviation across three independent runs.

Figure 9: Energy Consumption for different coefficient uncertainties on CIFAR-10. The shaded regions represent the standard deviation across three independent runs.

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
