# OpenReview forum: "Optimizing Federated Learning Client Selection via Multi-Objective Contextual Bandits"
_TMLR — Rejected by TMLR_

### Review · Reviewer_qxFY · 2025-07-31

**Summary Of Contributions:**

The authors propose a client selection framework in decentralized federated learning (FL) based on a contextual bandit approach, specifically leveraging a contextual zooming algorithm. The key goal is to balance model accuracy and energy efficiency by selecting clients that optimize a Pareto-efficient reward vector. This reward incorporates multiple objectives, but the authors focus primarily on accuracy and energy consumption.

**Audience:**

Yes

**Claims And Evidence:**

Yes

**Requested Changes:**

See above

**Strengths And Weaknesses:**

Strengths
Motivation and Relevance: The focus on energy-efficient federated learning is important. The paper rightly emphasizes that reducing energy consumption is crucial for real-world deployments of FL systems.

Novelty in Methodology: The use of a contextual zooming bandit algorithm is a nice methodological contribution. The idea of adaptively zooming into “critical” regions of the similarity space is well-motivated in the context of multi-objective client selection.

Multi-objective Formulation: The paper’s framing of client utility via Pareto-optimal tradeoffs between accuracy and energy efficiency is a principled approach to handling the competing goals of FL deployments.

Empirical Results: The claim that PCZFL “noticeably outperforms” state-of-the-art methods suggests strong empirical performance. Though the authors only consider CIFAR 10 which is small dataset, and the improvements aren't very pronounced, it seems like a good proof of concept for tmlr.

Weaknesses and Suggestions
Clarity in Writing:

The sentence “Energy-efficient clients not only reduce the frequency of battery charging but also minimize data transmissions, thereby resulting in lower overall energy consumption during model training” is somewhat awkward. The logic between reduced charging frequency and model training energy is indirect. Consider instead stating:

 “Energy efficiency—i.e., minimizing energy consumed per unit time of training—not only reduces battery strain but also cuts down on unnecessary data transmissions.”

The sentence “which adaptively ‘zooming in’ the critical regions…” is grammatically incorrect. It should read:

 “which adaptively zooms in on critical regions in the similarity space.”

Quantitative Claims in the Abstract:

 The statement “PCZFL noticeably outperforms current state-of-the-art methods…” would be more compelling if supported by a concrete metric in the abstract (e.g., “achieves 10% higher accuracy at 30% lower energy cost”).


Typographical Issue in Section 3:

 In the description of the arm set $Y = { y : y \in {0,1}^N, \sum_{i=1}^N y^i = 1 }$, the summation index i is incorrectly formatted as a superscript and should be a subscript.


Figure Clarity:


Figures 1 and 2 need better integration with the definitions in the text. From Definition 2, it is unclear how to interpret the layout or scaling of the “balls” in the similarity space.


The caption for Figure 2 should be more descriptive. Currently, it does not explain how the visual elements correspond to the reward formulation or contextual similarity.

---

> ### Author Response · Authors · 2025-09-17
> **Response to Reviewer qxFY**
>
> Comment: The sentence “Energy-efficient clients not only reduce the frequency of battery charging but also minimize data transmissions, thereby resulting in lower overall energy consumption during model training” is somewhat awkward. The logic between reduced charging frequency and model training energy is indirect. Consider instead stating: “Energy efficiency—i.e., minimizing energy consumed per unit time of training—not only reduces battery strain but also cuts down on unnecessary data transmissions.”
>
> Response: Thank you for pointing this out and we agree with your suggestion and have corrected it in the revised manuscript.
>
>
> Comment: The sentence “which adaptively ‘zooming in’ the critical regions…” is grammatically incorrect. It should read: “which adaptively zooms in on critical regions in the similarity space.”
>
> Response: Thank you for pointing this out and we agree with your suggestion and have corrected it in the revised manuscript.
>
>
> Comment: Quantitative Claims in the Abstract: The statement “PCZFL noticeably outperforms current state-of-the-art methods…” would be more compelling if supported by a concrete metric in the abstract (e.g., “achieves 10% higher accuracy at 30% lower energy cost”).
>
> Response: We appreciate the reviewer’s suggestion to strengthen the abstract with quantitative evidence. In the revised version, we have revised  the abstract accordingly. Specifically, we replaced the general statement “PCZFL noticeably outperforms current state-of-the-art methods” with the following: “In addition, we provide theoretical analysis on both the regret bound and time complexity of our method. Extensive experiments on CIFAR-10 demonstrate that PCZFL achieves a 4.7% improvement in global accuracy with 37.3% lower energy cost compared to Pow-d, the second-best in global accuracy, and delivers a 9.5% accuracy gain while reducing energy consumption by 30.9% relative to NCCB, the second-best in energy efficiency. On FMNIST, PCZFL further achieves a 3.2% improvement in global accuracy and a 21.4% reduction in energy cost relative to Pow-d, while delivering a 3.7% accuracy gain and a 30.8% reduction in energy consumption relative to NCCB.”
>
> Comment: Typographical Issue in Section 3: In the description of the arm set $\mathcal{Y}=\{y:y\in\{0,1\}^N,\sum_{i=1}^N y_i=1\}$, the summation index i is incorrectly formatted as a superscript and should be a subscript.
>
> Response: We thank the reviewer for pointing out the typographical issue in Section 3. We have corrected the arm set expression by replacing the summation index as a subscript. Furthermore, to avoid notational ambiguity, we now use $j$ to denote the objective dimension, thereby clearly distinguishing it from i, which continues to represent the clients index.
>
>
> Comment: Figure Clarity: Figures 1 and 2 need better integration with the definitions in the text. From Definition 2, it is unclear how to interpret the layout or scaling of the “balls” in the similarity space. The caption for Figure 2 should be more descriptive. Currently, it does not explain how the visual elements correspond to the reward formulation or contextual similarity.
>
> Response: We have augmented the captions for Figures 1 and 2 to provide a more detailed explanation of how Definition 2 is reflected in the visual illustrations.

---

> ### Author Response · Authors · 2025-09-25
>
> Thank you again for your valuable comments on our manuscript. We have revised the paper to address your suggestions, including improvements to the wording and grammar, clarifications in the abstract, corrections to notation, and more detailed figure captions.
>
> We believe these revisions have addressed your concerns. Please let us know if there are any remaining points we may have overlooked or if further adjustments are needed.
>
> We sincerely appreciate your constructive feedback and the time you’ve taken to help us improve this work.

---

### Review · Reviewer_CbH4 · 2025-08-12

**Summary Of Contributions:**

The paper introduces PCZFL, a multi-objective contextual bandit approach to FL client selection that optimizes both accuracy and energy efficiency by grouping clients in a similarity space and selecting those on the Pareto front, showing empirical gains over baselines on CIFAR-10.

**Audience:**

Yes

**Claims And Evidence:**

Yes

**Requested Changes:**

See my comments above.

**Strengths And Weaknesses:**

Strengths:

1. Addresses a practically relevant dual-objective problem in FL (accuracy and energy efficiency) with a clear formulation using multi-objective contextual bandits.

2. Provides thorough experimental comparisons against multiple baselines, with visualizations that help illustrate the algorithm’s behaviour in the accuracy-energy trade-off space.


Weakness

1. The experimental setup assumes that each client has a random but fixed per-round energy consumption throughout training. While this simplifies analysis and isolates the algorithm’s effect, it does not reflect real-world FL environments, where energy usage can vary significantly between rounds due to changing model sizes, data heterogeneity, device workloads, and network conditions. As a result, the reported energy savings may overstate the stability of PCZFL’s advantage in practice. It would strengthen the work to either (1) justify why the fixed-energy assumption is appropriate for the targeted deployment scenarios, or (2) include experiments with more realistic, time-varying energy consumption models following prior measurement-based studies.



2. While the paper provides a regret bound for the proposed client selection policy (adapted from Turgay et al., 2018), it does not establish any formal convergence rate for the federated learning process itself. In practice, a low-regret selection policy does not automatically guarantee faster or more stable convergence of the global model—especially in non-i.i.d., multi-objective FL settings where partial participation and client drift can have significant effects. Without such an analysis, it is unclear whether PCZFL’s gains in empirical accuracy reflect inherently better convergence behavior or are artifacts of the experimental setup. This omission weakens the theoretical contribution and leaves a critical gap in understanding the method’s performance guarantees.

3. The dataset used for the experiments is very simple. With such limited experiments, I cannot recommend acceptance of this paper.

---

> ### Author Response · Authors · 2025-09-17
> **Response to Reviewer CbH4**
>
> Comment: The experimental setup assumes that each client has a random but fixed per-round energy consumption throughout training. While this simplifies analysis and isolates the algorithm’s effect, it does not reflect real-world FL environments, where energy usage can vary significantly between rounds due to changing model sizes, data heterogeneity, device workloads, and network conditions. As a result, the reported energy savings may overstate the stability of PCZFL’s advantage in practice. It would strengthen the work to either (1) justify why the fixed-energy assumption is appropriate for the targeted deployment scenarios, or (2) include experiments with more realistic, time-varying energy consumption models following prior measurement-based studies.
>
> Response: We appreciate the reviewer’s thoughtful observation. In our experiments, we adopted a random but fixed per-round energy consumption model primarily to focus the analysis on the effect of the client selection strategy itself. This assumption enabled us to clearly highlight the relative trade-offs between accuracy and energy efficiency more clearly, without introducing additional confounding factors.
> That said, we emphasize that PCZFL’s client selection policy inherently operates on contextual information available in each round, which includes measures related to both accuracy and energy. Therefore, even if energy consumption were time-varying, as in more realistic FL deployments, the algorithm would naturally adapt its selections based on the updated context at each round.
> We agree that incorporating more realistic, time-varying energy consumption models, such as those derived from prior measurement-based studies, would further strengthen our evaluation and provide additional empirical validation. To address this point, we will add a note in the discussion section to clarify this modeling choice and explicitly point to such extensions as a valuable direction for future work.
>
>
> Comment: While the paper provides a regret bound for the proposed client selection policy (adapted from Turgay et al., 2018), it does not establish any formal convergence rate for the federated learning process itself. In practice, a low-regret selection policy does not automatically guarantee faster or more stable convergence of the global model—especially in non-i.i.d., multi-objective FL settings where partial participation and client drift can have significant effects. Without such an analysis, it is unclear whether PCZFL’s gains in empirical accuracy reflect inherently better convergence behavior or are artifacts of the experimental setup. This omission weakens the theoretical contribution and leaves a critical gap in understanding the method’s performance guarantees.
>
> Response: We thank the reviewer for this insightful comment. Our primary focus in this work is to introduce a mapping of the Pareto Contextual Zooming (PCZ) framework into the federated learning setting in order to address multi-objective goals such as accuracy and energy efficiency. Accordingly, our theoretical contribution has been centered on establishing the regret bound of the proposed client selection policy, which directly characterizes the trade-offs in client selection under multi-objective constraints.
> We agree that providing a formal convergence rate analysis of the global federated learning process under partial participation and non-i.i.d. settings would further strengthen the theoretical guarantees. However, such an analysis is non-trivial and beyond the immediate scope of this paper, since it requires new assumptions and proof techniques to bridge the regret of client selection with the convergence behavior of the federated model itself. We view this as an important and valuable direction for future work, and we will highlight it explicitly in the revised version.
>
> Comment: The dataset used for the experiments is very simple.
>
> Response: Due to the time and resource constraints, we were only able to expand our experiments by incorporating the Fashion-MNIST (FMNIST) dataset in the revised paper. FMNIST provides a complementary perspective by exhibiting different convergence and stability behaviors in federated settings. For example, FMNIST highlights how PCZFL adapts under conditions of faster convergence, which contrasts with the more gradual dynamics of CIFAR-10. We acknowledge that both datasets remain limited compared to large-scale, real-world FL applications. However, our choice of CIFAR-10 and FMNIST aligns with established practice in the federated learning literature, where they are widely adopted for benchmarking client selection and system-level algorithms (e.g., Pow-d, DivFL). These benchmarks allow for reproducibility and fair comparison against prior work. Looking forward, we plan to extend our evaluation for future work to more complex and large-scale datasets such as CIFAR-100, REDDIT, and LEAF benchmarks, which more closely mimic real-world heterogeneity.

---

> ### Author Response · Authors · 2025-09-25
>
> Thank you again for your valuable and constructive feedback regarding the energy consumption assumptions, the lack of formal convergence analysis, and the simplicity of the experimental setup. We took these concerns very seriously and have carefully addressed these points in our revision through both substantial clarifications in the text and additional experiments and analyses.
>
> We believe these revisions have addressed your concerns. Please let us know if further improvement is required.
>
> We greatly appreciate the time and effort you invested in providing such valuable feedback.

---

### Review · Reviewer_uQSC · 2025-09-07

**Summary Of Contributions:**

This paper addresses the challenge of the federated learning (FL) problem in scenarios where clients have multi-objective constraints like model accuracy and resource constraints. The authors propose a novel client selection mechanism that balances multiple objectives. The proposed method leverages a multi-objective contextual bandit framework to dynamically select clients based on their context similarity, which includes model accuracy and energy consumption. The key idea is to find a Pareto-optimal set of clients that can contribute to improving the multi-objective performance of the global model. The selection strategy is designed to adaptively explore the client space while exploiting the knowledge gained from previous selections. PCZFL employs a zooming technique to focus on the Parato-optimal subset of clients by partitioning the context space into smaller balls and selecting clients from the relevant balls based on their upper confidence bounds.

The authors provide experimental results based on CIFAR-10 datasets to demonstrate the effectiveness of their approach in improving the overall performance of the federated learning process while reducing energy consumption.

**Audience:**

No

**Broader Impact Concerns:**

This work is highly based on the Pareto Contextual Zooming (PCZ) algorithm proposed by Turgay et al.[1].
Turgay et al. formulated the multi‐objective contextual bandit problem with theoretical regret bounds in abstract similarity spaces. PCZFL instantiates this framework for federated learning scenarios. I found that the upper confidence bound and the updating rule of balls in PCZFL are the same as those of PCZ. Also, the distance function defined in PCZFL is Euclidean distance, while the distance function defined in PCZ is more general. Therefore, I have a concern about the novelty of PCZFL compared to PCZ, especially since PCZFL lacks a theoretical analysis of regret bounds.

[1] Turgay, E., Oner, D., & Tekin, C. (2018, March). *Multi-objective contextual bandit problem with similarity information*.

**Claims And Evidence:**

Yes

**Requested Changes:**

I list several points that need to be addressed to improve the clarity and quality of the paper:

* The definition of the domain of Ball (Definition 2) has an inconsistency. A time step notation $t$ appears in the LHS but does not appear on the RHS. It is better to use $\mathcal{B}\_t$ and $\mathcal{B}_{t+1}$ instead of adding $t$ in $\text{dom}_t(\cdot)$.
* The font size of Figures 1 and 2 is too small to read.
* I did not see notation $B'$ in Figures 1 and 2. Although I understand that the authors claim that $B'$ is any ball contained in the Ball $B$, it would be better to explicitly show $B'$ in the figures or ignore $B'$ in the description of the figures.
* In the histogram plot in Figures 1 and 2, including the notation defined in Eq. (4) of each term in the legend will be better for understanding.
* $D(B, B')$ represents the distance between the centers of $B$ and $B'$, but it is not aligned with the arrow in Figures 1 and 2 where they are the distance between the edges of $B$ and $B'$.
* In step 8 of the PCZFL, the algorithm needs to select all clients from the domain of every ball in $\mathcal{B}$. The ball $B$ seems to be a set of $X^T y$ while $X$ is a client's context and $y$ is a selector vector. However, there is no clear mathematical definition of the selection process. It would be better to define the selection process mathematically.
* In the experiments, the authors should include more details about the experimental setup, such as how the clients are simulated, what the difference is between the clients, and how many runs are averaged to produce the results. It would also be helpful to provide more information about how to calculate the performance metrics used in the experiments.
* In Figures 4, 5, 6, and 7, what does the shaded area represent? It would be better to explain it in the caption.

**Strengths And Weaknesses:**

## Strengths:

The paper introduces an innovative application of Pareto Contextual Zooming to federated learning client selection, extending multi-objective contextual bandits to address the dual challenges of accuracy optimization and energy efficiency. The formulation of client selection as a contextual MAB problem with multiple objectives represents a meaningful theoretical contribution that bridges bandit algorithms and federated learning.

The author includes a detailed algorithmic description of the proposed PCZFL method and a computational complexity analysis, which enhances the clarity and reproducibility of the work. The experimental results demonstrate the effectiveness of PCZFL in improving model accuracy while reducing energy consumption.

## Weaknesses:

The paper claims that PCZFL adapts to the dynamic contexts of clients. Usually, the dynamic context means that the context of each client changes over time. If this is the case, the authors need to clarify how PCZFL handles the time-varying nature of client contexts and include experimental results that demonstrate the algorithm's performance in such scenarios. However, if the context of each client is static and does not change over time, the authors should clarify this point in the paper.

The experimental section lacks sufficient details about the experimental setup, such as how clients are simulated, the differences between clients, and how energy consumption is measured. Providing more information about the performance metrics used in the experiments would enhance the credibility of the results.

---

> ### Author Response · Authors · 2025-09-17
> **Response to Reviewer uQSC**
>
> Comment: The definition of the domain of Ball (Definition 2) has an inconsistency. A time step notation t appears in the LHS but does not appear on the RHS. It is better to use B_t and B_{t+1} instead of adding t in dom_t(·).
>
> Response: We thank the reviewer for pointing out this inconsistency in the definition of the domain of a ball. To address this concern, we have revised the notation by replacing the dependency on time step $t$ to the ball set $\mathcal{B}$, which is the set of all existing balls at the current moment. Since the domain of a ball depends on the collection of active balls rather than on an explicit time index, we believe this revised notation is more precise and eliminates the ambiguity.
>
> Comment: The font size of Figures 1 and 2 is too small to read.
>
> Response: We have increased the font size in Figures 1 and 2, and enlarged the overall figure dimensions in the revised manuscript to improve readability and presentation.
>
> Comment: I did not see notation B’ in Figures 1 and 2. Although I understand that the authors claim that B’ is any ball contained in the Ball B, it would be better to explicitly show B’ in the figures or ignore B’ in the description of the figures.
>
> Response: We thank the reviewer for raising this point. We would like to clarify that $B'$ is not restricted to being a ball contained in $B$; rather, $B'$ can be any ball, including $B$ itself (see Eq.(5)). The chosen $B'$ is determined by evaluating all combinations of $g_{B'}^{j,\text{UCB}}$ and $D(B',B)$. It appears in the right subfigure of Figure 1. To avoid misunderstanding, we have kept $B'$ in the figure captions and added the following explanatory line to make this explicit:
> Every ball $B$ will need to identify a related ball $B'$ with the minimum combining score of $g_{B'}^{j, \text{UCB}}$ and $D(B', B)$
> We believe this clarification preserves the correctness of the definition and prevents ambiguity regarding the role of $B'$.
>
> Comment: In the histogram plot in Figures 1 and 2, including the notation defined in Eq. (4) of each term in the legend will be better for understanding.
>
> Response: We thank the reviewer for this helpful suggestion. In the revised manuscript, we have adjusted the labels of the histograms in Figures 1 and 2 so that they directly match the notation defined in Eq. (4), which improves consistency and clarity.
>
> Comment: D(B,B’) represents the distance between the centers of B and B’, but it is not aligned with the arrow in Figures 1 and 2 where they are the distance between the edges of B and B’.
>
> Response: We thank the reviewer for pointing out this inconsistency. In the revised manuscript, we have adjusted the arrow lines in Figures 1 and 2 so that they correctly point to the centers of the balls, consistent with the definition of $D(B, B')$ as the distance measured between the centers of $B$ and $B'$.
>
> Comment: In step 8 of the PCZFL, the algorithm needs to select all clients from the domain of every ball in B. The ball B seems to be a set of $X^\top y$ while X is a client’s context and y is a selector vector. However, there is no clear mathematical definition of the selection process. It would be better to define the selection process mathematically.
>
> Response: We thank the reviewer for this helpful suggestion. To address this concern, we revised step 8 of the algorithm, mathematically defining the chosen clients.
>
> Comment: In the experiments, the authors should include more details about the experimental setup, such as how the clients are simulated, what the difference is between the clients, and how many runs are averaged to produce the results. It would also be helpful to provide more information about how to calculate the performance metrics used in the experiments.
>
> Response: We thank the reviewer for this valuable comment. In the revised manuscript, we have expanded the description of the experimental setup to clarify how clients are simulated, how they differ, and how the performance metrics are calculated, using the following: “Specifically, we set the Dirichlet concentration parameter to $\alpha = 0.1$, which results in highly skewed local class distributions across clients and thus simulates strong heterogeneity in the federation. For evaluation, we focus on two key metrics: (i) global accuracy, defined as the test accuracy of the aggregated global model, and (ii) accumulated energy consumption, calculated as the sum of the per-round energy consumption of all selected clients across the entire training process, where lower values indicate improved energy efficiency. ”
>
> Regarding the number of runs, it is provided at the beginning of Section 5, where we state: “We averaged the results over three runs to improve the reliability of our findings.”

---

> ### Author Response · Authors · 2025-09-17
> **Response to Reviewer uQSC (Continued)**
>
> Comment: In Figures 4, 5, 6, and 7, what does the shaded area represent? It would be better to explain it in the caption.
>
> Response: We thank the reviewer for this helpful suggestion. In the revised manuscript, we have clarified the meaning of the shaded regions in Figures 4, 5, 6, and 7 by adding the following explanation to each caption: “The shaded regions represent the standard deviation across three independent runs.”
>
>
> Broader Impact comment: This work is highly based on the Pareto Contextual Zooming (PCZ) algorithm proposed by Turgay et al.[1]. Turgay et al. formulated the multi‐objective contextual bandit problem with theoretical regret bounds in abstract similarity spaces. PCZFL instantiates this framework for federated learning scenarios. I found that the upper confidence bound and the updating rule of balls in PCZFL are the same as those of PCZ. Also, the distance function defined in PCZFL is Euclidean distance, while the distance function defined in PCZ is more general. Therefore, I have a concern about the novelty of PCZFL compared to PCZ, especially since PCZFL lacks a theoretical analysis of regret bounds.
>
> Response: We thank the reviewer for this important comment. Our work is indeed inspired by the Pareto Contextual Zooming (PCZ) algorithm of Turgay et al. [1]. The novelty of our contribution lies in instantiating and adapting this framework to the federated learning (FL) setting, where client selection must simultaneously optimize global accuracy and energy efficiency under partial participation and the presence of non-IID data distributions. While PCZ provides a general multi-objective contextual bandit formulation, our PCZFL introduces several non-trivial adaptations: (i) the context is redefined to capture client-specific accuracy and energy characteristics, coupled with a nontrivial design of similarity space; (ii) the ball domain is mapped to federated clients to enable selection under heterogeneous FL conditions; and (iii) the framework is implemented and evaluated in real-world FL settings, demonstrating how Pareto contextual zooming can explicitly balance accuracy and energy in practice.
> Regarding the theoretical contribution, we provide a regret bound for the client selection policy adapted to FL. We acknowledge that a full convergence analysis of the federated optimization process itself is not included, and we consider this a valuable direction for future work.
>
> We believe that PCZFL is not simply a restatement of PCZ, but a concrete adaptation and extension to FL that addresses practical challenges such as non-IID data, communication and energy constraints, and partial participation, supported by extensive empirical validation.

---

> ### Author Response · Authors · 2025-09-25
>
> Thank you again for your detailed and constructive feedback on our manuscript. We have carefully revised the paper to address your comments, including clarifications to definitions and notation, improvements to figure readability and consistency, a more precise description of the client selection process, and expanded details on the experimental setup and evaluation metrics. We also clarified the novelty of PCZFL relative to PCZ and highlighted future directions regarding theoretical analysis.
>
> We believe these revisions have addressed your concerns. Please let us know if there are any aspects we may have overlooked or where further adjustments would be helpful.
>
> We greatly appreciate the time and effort you dedicated to strengthening this work.

---

### Decision · Action_Editor_V1qy · 2025-10-23

**Recommendation:** Reject

**Audience:**

Yes

**Audience Explanation:**

This submission should be of interest to audience in multi-objective objective, bandits, and federated learning.

**Claims And Evidence:**

No

**Claims Explanation:**

This paper proposes a multi-objective contextual bandit client selection framework that directly uses prior work (Turgay et al. 2018). As reviewers pointed out, no additional challenges are mentioned in using it in the context of federated learning. In addition, the paper presents regret guarantees following Turgay et al. (2018), but does not mention or address the gap between the multi-armed bandit problem and the client selection problem. For instance, one critical difference is that prior regret analysis focuses on the setting where one arm is selected at each iteration, where this work selects multiple clients and does some aggregation of the models. Given the difference, previous bound may not be directly used. No proof is given for statements in Section 4.3. Experiment results are mixed and conducted on small-scale datasets. Writing can be further improved by putting the technical contextual bandit problem into the FL application in a more accessible way.

**Resubmission Of Major Revision:**

The authors may consider submitting a major revision at a later time.